# DEEP Q LEARNING FROM DYNAMIC DEMONSTRATION WITH BEHAVIORAL CLONING

## ABSTRACT

Although Deep Reinforcement Learning (DRL) has proven its capability to learn optimal policies by directly interacting with simulation environments, how to combine DRL with supervised learning and leverage additional knowledge to assist the DRL agent effectively still remains difficult. This study proposes a novel approach integrating deep Q learning from dynamic demonstrations with a behavioral cloning model (DQfDD-BC), which includes a supervised learning technique of instructing a DRL model to enhance its performance. Specifically, the DQfDD-BC model leverages historical demonstrations to pre-train a supervised BC model and consistently update it by learning the dynamically updated demonstrations. Then the DQfDD-BC model manages the sample complexity by exploiting both the historical and generated demonstrations. An expert loss function is designed to compare actions generated by the DRL model with those obtained from the BC model to provide advantageous guidance for policy improvements. Experimental results in several OpenAI Gym environments show that the proposed approach adapts to different performance levels of demonstrations, and meanwhile, accelerates the learning processes. As illustrated in an ablation study, the dynamic demonstration and expert loss mechanisms with the utilization of a BC model contribute to improving the learning convergence performance compared with the origin DQfD model.

## 1 INTRODUCTION

Deep reinforcement learning (DRL) methods have made great progress (Mnih et al., 2013; 2015; Silver et al., 2017) when applied in several rule-based applications such as the Go game (Silver et al., 2016). However, due to the diversity and uncertainty of complex systems, the establishment of a simulation environment is difficult to be consistent with the real-world system. Therefore, DRL algorithms usually fail in a direct application to many real-world scenarios. Meanwhile, a DRL model may produce an action that is sampled from a random policy when exploring the state-action space. However, random actions are not allowed in many real-world circumstances. For example, in autonomous driving experiments (Kiran et al., 2020), a random policy may contribute to traffic congestion, even road accidents. Therefore, fitting to complex situations becomes one of the most urgent tasks for applying a DRL model for complicated decision-making tasks.

It is noted that human experts have great advantages in learning efficiency and decision-making performance (Tsividis et al., 2017). Incorporating expert knowledge is a potential solution to enhance the adaptability of DRL models for complex tasks (Hester et al., 2018; Matas et al., 2018). Nevertheless, the knowledge and experience of an expert are difficult to be modeled and described directly. One solution, attracting more and more attention, is to indirectly learn expert strategies by learning their decision trajectories, also known as demonstrations (Schaal, 1997; Behbahani et al., 2019; Ravichandar et al., 2020). Particularly, deep Q learning from demonstrations (DQfD) is a typical algorithm that succeeded in combining DRL with demonstrations (Hester et al., 2018), which combines the temporal difference (TD) error in the traditional DDQN algorithm with supervised expert loss by constructing a hybrid loss function. Through a specially designed large margin supervised loss function (Piot et al., 2014a;b), the DQfD method can guide and assist an agent to learn the expert's knowledge by constantly steering the agent learning strategies closer to those represented by the demonstration.

However, the DQfD model suffers from three major issues: (1) In the DQfD learning process, the trajectory data in the historical demonstration dataset is the only source for contributing expert loss values, which does not include the self-generated transitions of the trained agents. As a result, the DQfD algorithm merely relies on TD errors to improve the policy but the demonstration is idle when the self-generated transitions are sampled from the experience replay buffer, which reduces the efficiency of utilizing demonstrations. (2) According to the learning mechanism, static demonstrations are too limited to cover sufficient state-action space during the agent training process, especially when it is difficult or expensive to collect demonstrations in real-world applications.

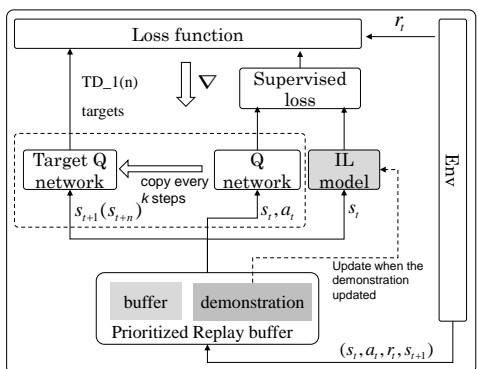

Figure 1: The technique framework of the proposed DQfDD-BC method where IL model represents an imitation learning model.

Also, with more newly-generated transitions added into the experience replay buffer, historical demonstrations would make smaller contributions to the policy improvement as their sampling probability becomes lower and lower. (3) The DQfD algorithm requires the learned policy to approximate the demonstration but ignores the imperfection of historical demonstrations and it is an imperfect demonstration that commonly exists in real-world applications. Perfect demonstration can always provide an appropriate guide but the imperfect demonstration is detrimental to the improvement of the policy when the learned policy is superior to the policy represented by the imperfect demonstration.

To address the above issues, we propose a novel deep Q learning from dynamic demonstration with the behavioral cloning (DQfDD-BC) method. Fig. 1 illustrates the structure of the proposed approach. DQfDD-BC shares the same basic components of the DDQN algorithm (Hasselt et al., 2016), including two Q networks and an experience replay buffer. There are an additional imitation learning (IL) (Hussein et al., 2017) module and an alterable demonstration in the replay buffer. Besides, the loss value is calculated considering both the output of the trained Q network and the BC model. Two main contributions are summarized as follows:

(1) An IL model, named behavioral cloning (BC) (Torabi et al., 2018; Bühler et al., 2020), is proposed to generate expert loss to utilize all the transitions in the experience replay buffer. It first attempts to extract the experts' policy from the initial demonstration and allows the agent to provide reasonable actions when facing newly generated states. During the self-learning process, the agent's actions are compared with those generated by the BC model through a self-designed expert loss function. The inclusion of the BC model allows the knowledge in the demonstrations to be sufficiently utilized in the training process and enables the model to cope with the states which experts have not ever encountered. The supervised learning process and self-learning process can promote each other. The supervised model provides a basic reference to guide the model adjustment. Meanwhile, the self-learning process keeps improving and overcomes the limitation of the BC model and suboptimal samples during the learning process.

(2) An automatic update mechanism is proposed to adaptively enhance the BC model. In particular, new transitions are generated by the trained agents to fine-tune the BC model if the model achieves a relatively high-performance score. Such a mechanism tries to include more high-quality transition samples to improve the demonstration and avoids potential adverse impacts caused by imperfect demonstrations.

In this study, we evaluate the proposed DQfDD-BC method on several gym environments (Brockman et al., 2016). For comparison purposes, the DDQN and DQfD methods are used as the baselines (Hasselt et al., 2016; Hester et al., 2018). The experiments clearly demonstrate that DQfDD-BC surpasses all the baselines in terms of convergence speed as well as decision-making performance. The ablation experiments also show that both the proposed expert loss function together with the BC model and dynamic demonstration contribute significantly to the performance superior to the DQfDD-BC algorithm.

## 2 RELATED WORKS

In most cases, demonstration comes from the human or other kinds of "experts", who can provide valuable information for different hard problems, such as robot control (Schaal, 1997; Ravichandar et al., 2020), self-driving (Bojarski et al., 2016) and so on. As a kind of approaches which can make use of the demonstration, behavioral cloning (Torabi et al., 2018), one of imitation learning (Hussein et al., 2017; Brown et al., 2020), has received extensive attention (Zhang et al., 2018; Bühler et al., 2020) due to its significant advantages, such as fast learning speed, simplicity, and effective utilization of demonstration and so on. It constructs mappings from states to actions (or distributions of actions) in a supervised manner to model the policies represented by the demonstrations and achieves the goal of learning from a demonstration by minimizing various supervised losses. In a study by NVIDIA, the model obtained interesting results by minimizing the mean squared error between the steering command output by the network and the command of a human driver (Bojarski et al., 2016). *Chauffeurnet* further enhances the performance of imitation learning in autopilot by synthesizing the worst scenarios (Bansal et al., 2018).

DAgger (Ross et al., 2011) is proposed to cope with the covariate shift problem and experts are required to respond to self-generated states so that totally new transitions are generated and they can expand the state-action space covered by demonstration. The goal of performance improvement can be achieved by adding new samples continuously during the learning process. However, the DAgger method requires an always-available expert to assist in labeling the data, which reduces the practicality of the method. Deeply AggreVaTeD (Sun et al., 2017) enables the DAgger to handle continuous action spaces by using deep neural networks but the weakness of DAgger was also preserved.

Another typical class of imitation learning algorithms is generative adversarial imitation learning (GAIL) (Ho & Ermon, 2016; Wang et al., 2019). Instead of mapping from state to action directly, GAIL learns from the demonstrations by introducing the generator and discriminator. The generator is used to generate state-action pairs and learn a policy, while the discriminator is used to distinguish whether a state-action pair is from the expert or the learned policy. The information contained in the demonstration is indirectly combined with policy improvement through the process of adversarial learning. GAIL shows good performance on the high-dimensional continuous-control problem (Kang et al., 2018; Song et al., 2018). Policy optimization with demonstrations (POfD) (Kang et al., 2018) utilizes the demonstration by an adversarial learning method and has made efforts on sparse and dense reward environments.

Recently, demonstrations have been leveraged to tutor DRL to achieve better performance. Since the DRL reward is immediate feedback from the environment for evaluating the performance of the policy, reshaping the reward function through the demonstration is an effective approach to improve DRL performance (Brys et al., 2015; Suay et al., 2016). By transforming the form of the reward function, the difficulty of agent training has been reduced. Model-Based DRL (MBRL) has shown great power in some difficult tasks (Kaiser et al., 2019) and demonstration is also leveraged to deal with the MBRL problems (Lambert et al., 2019; Thananjeyan et al., 2020).

All of the above methods yielded impressive results but most of them utilized demonstrations through reward shaping or imitating the demonstration. Another idea is to learn from demonstrations directly to improve the policy. The transitions in the experience replay buffer and demonstrations are both extracted by DRL agents (Hester et al., 2018; Oh et al., 2018; Xiong et al., 2019). These approaches put the demonstration into the experience replay buffer and sample from the hybrid experience replay buffer. The DQfD algorithm enhances the DDQN (Hasselt et al., 2016) algorithm by keeping the demonstration in the replay buffer at all times. A designed supervised expert loss is generated when the transitions in the demonstration are sampled to update the parameters of the neural networks. The self-imitation learning (SIL) method chooses different loss functions and adds new trajectories to the demonstration. It updates the same neural network twice with the A2C loss function and the SIL loss function (Oh et al., 2018). The DDPGfD algorithm (Vecerik et al., 2017) is similar to the DQfD and they both assist in enhancing the original algorithms through a hybrid loss function with the "expert loss". The difference between the two is that the DDPGfD algorithm is based on the DDPG algorithm (Lillicrap et al., 2015), which is designed to solve continuous action space problems. However, both DQfD and DDPGfD suffer from the problem of under-exploiting demonstration data (Kang et al., 2018) and our DQfDD-BC method has made new and comprehensive progress on solving the problem.

## 3 PROBLEM STATEMENT

We consider the standard Markov Decision Process (MDP) (Sutton & Barto, 2018), which is defined by a tuple $\langle \mathcal{S}, \mathcal{A}, \mathcal{P}, r, \gamma \rangle$, where $\mathcal{S}, \mathcal{A}, \mathcal{P}, r, \gamma$ represent the state space, action space, state transition distribution, reward function and discount factor, respectively. The transition distribution $\mathcal{P}(s' \mid s, a)$ describes the process of taking action $a$ at state $s$, $s'$ is the next state and reward function $r(s, a)$ gives the feedback. A policy $\pi(a|s)$ specifies how the agent responses to each state. The goal of the agent is to find the policy $\pi$ whcih maps states to actions that maximizes the expected discounted total reward $\mathbb{E}_\pi \left[ \sum_{t=0}^{\infty} \gamma^t r(s_t, a_t) \right]$ over the agent's lifetime.

In the DQfD algorithm, the complete loss (Eq. 1) function contains four parts: $J_{DQ}(Q)$, $J_n(Q)$, $J_E(Q)$ and $J_{L2}(Q)$, which are the double Q learning TD loss, n-step double Q learning TD loss, expert loss and L2 regularization loss, respectively. The $\lambda$ parameters control the weighting between the losses. The expert loss (Eq. 2) in DQfD is the most significant loss among the losses, allows the model to satisfy the Bellman equation, and also enables the agent to learn from the demonstrations. The large margin supervised loss $l(a_E, a)$ is 0 when $a = a_E$ with $a_E$ being the expert action in the demonstration, and a positive constant otherwise. This loss forces the Q values of the other actions to be at least a margin lower than the value of the demonstrator's action (Hester et al., 2018). It is noted that, when the sampled transitions are not in the demonstration, $\lambda_2 = 0$ and the expert loss $J_E(Q)$ is non-functional.

$$J(Q) = J_{DQ}(Q) + \lambda_1 J_n(Q) + \lambda_2 J_E(Q) + \lambda_3 J_{L2}(Q) \tag{1}$$
$$J_E(Q) = \max_{a \in A} \left[ Q(s, a) + l(a_E, a) \right] - Q(s, a_E) \tag{2}$$

## 4 METHODOLOGY

### 4.1 DYNAMIC DEMONSTRATIONS FOR DQFD

As the transition quality of historical demonstrations is usually much higher than that associated with random policies, a DQfD agent learns its initial knowledge from preset demonstrations. One of the major improvements of our proposed DQfDD-BC approach is adding the newly generated interactions into the demonstrations. These newly generated transitions, who have a higher performance score for the same task, are collected from a trained model. Particularly, after the early stage of the training process, the transition samples are automatically generated and inserted into the demonstration dataset when the final cumulative reward of each episode reaches a new high score. By adding new and better transition samples after the performance of the model has been improved effectively, the quality of the transition samples is continuously improved while attempting to cover the full state-action space. Such a mechanism can continuously improve the performance of the policy represented by the demonstration, and generate a continuously positive effect on the model performance.

The idea of the proposed approach is close to the DAgger algorithm, especially for improving the model's decision-making capability by adding new transitions to the experience replay buffer and continuously using the added data to optimize the model parameters. However, unlike DAgger, our method does not rely on manual labeling and automatically determines whether new data need to be added to the demonstration, which can significantly reduce the computationally cost and improves the applicability of the method.

### 4.2 EXPERT LOSS WITH SUPERVISED BC MODEL

The DQfD method originally leverages a binary supervised large margin loss function $l(a_E, a)$ to compare the generated actions with those in demonstrations under the same environment states. However, the values of the large margin loss function are prone to cause a volatile gradient and lead to the instability of the training process. In this study, DQfDD-BC is designed for discrete action spaces such that the BC model is considered to solve a multi-label classification problem.

In detail, the proposed DQfDD-BC method contains a deep neural network-based BC model and maintains two different experience replay buffers: $D^{replay}$ and $D^{demo}$. In particular, $D^{replay}$ refers

to the common experience replay buffer in a DRL model while $D^{demo}$ consisting of both historical and self-generated demonstrations. The BC model is first pre-trained with demonstrations $D^{demo}$ to obtain its initial decision-making ability. Prioritized experience replay mechanism (Schaul et al., 2015) is also applied to both of the two replay buffers to improve the sample efficiency. The performance of the model improves all the time during the learning process, so newly generated transition has always been given the highest priority to ensure that transitions with higher performance are extracted timely, and the value of complete loss function is used to update the priority of the transition samples.

The DQfDD-BC model takes full advantage of the demonstration by generating expert losses for all self-generated transitions instead of directly using historical demonstrations. Eq. 3 is used to pre-train the BC model and Eq. 4 evaluates the difference between $a$ and $\pi_{bc}(s)$, where $a$ is the action provided by the Q network and $\pi_{bc}(s)$ is the action generated by the latest BC model. In Eqs. 3 and 4, $a_E$ refers to the action in the demonstration and $\pi_{bc}$ represents the learned policy of the BC model. The supervised loss $l(a, \pi_{bc}(s))$ is zero when the action of the Q network output is the same as the output of the BC model and otherwise a positive number. Consequently, the complete expert loss function of DQfDD-BC is shown as Eq. 5.

$$l_{BC} = Crossentropy(a_E, \pi_{bc}(s_t)) \tag{3}$$

$$l(a, \pi_{bc}(s)) = Crossentropy(a, \pi_{bc}(s)) \tag{4}$$

$$J_E(Q) = \max_{a \in A} [Q(s, a) + l(a, \pi_{bc}(s))] - Q(s, a_E) \tag{5}$$

---

**Algorithm 1:** DQfDD-BC: Deep Q learning from Dynamic Demonstration with Behavioral Cloning

---

1   **Initialization**: $D^{replay}$ and $D^{demo}$: both initialized with the incipient demonstration data set,

    $\theta$: weights for initial Q network, $\theta'$: weights for the target network, $\tau$: frequency at which to update the target network, $E$: maximum number of episodes, $M$: maximum number of newly generated trajectories, $k$: number of pre-training gradient updates;

2   Train the BC model with $D^{demo}$ with the behavioral cloning cross-entropy loss (Eq. 3);

3   **for** *step = 1* **to** $k$ **do**

4      Sample a mini-batch of transitions from $D^{replay}$ with prioritization;

5      Calculate expert loss $J_E(Q)$ using the BC model;

6      Calculate loss $J(Q)$ using target net;

7      Perform a gradient descent step to update $\theta$;

8      every $\tau$ steps update the target network $\theta' = \theta$;

9   **for** *episode = 1* **to** $E$ **do**

10     **while** *not done* **do**

11        Sample action $a$ from $\epsilon - greedy$ policy;

12        Play action $a$ and observe $(s', r, done)$;

13        Store $(s, a, r, s', done)$ into $D^{replay}$;

14        Sample a mini-batch of transitions from $D^{replay}$ with prioritization;

15        Calculate expert loss$J_E(Q)$ using the BC model;

16        Calculate $J(Q)$ using target net;

17        Perform a gradient descent step to update $\theta$;

18        every $\tau$ steps update the target network $\theta' = \theta$;

19        $s = s'$;

20     **if** *get the best episode score* **then**

21        **for** *i = 1* **to** $min(episode//2, M)$ **do**

22           Agent interacts with the environment until $done$;

23           Store the trajectory into $D^{demo}$;

24        Fine-tune the BC model with new $D^{demo}$;

---

This expert loss function aims to smooth the output of the loss function and stabilize the training process. More importantly, the BC model can be used to provide reasonable actions and generate supervised losses for all self-generated transitions in the experience replay buffer. Compared with the DQfD method, the sample efficiency is significantly improved.

The pseudo-code of the proposed DQfDD-BC method is sketched in Algorithm 1 and consists of three main stages: BC model pre-training, agent pre-training, and joint model self-learning. Firstly, the BC model is pre-trained with Eq. 3 to gain the initial decision-making ability, and then the Q network is also pre-trained in a supervised manner. Through the trained BC model, the information in the demonstrations indirectly contributes to the improvement of the agent's performance. During the self-learning process, the trained BC model is leveraged to generate an expert loss to construct the complete loss function. Combined with TD errors, the self-learning ability is retained and the L2 regularization losses on the network weights to prevent over-fitting. After each episode, we evaluate the policy and apply the dynamic demonstration method to $D^{demo}$ to update the demonstration. The best episode score is taken from the final cumulative reward of each episode compared to the previous history. To reduce the influence of the lucky episode, the demonstration will not be updated in the early stage of the self-learning process. We begin to update the demonstration after the self-learning process has last for at least 10 episodes. Fine-tuning of the BC model is conducted after $D^{demo}$ is updated, thus enabling the BC model to constantly improve the supervised strategy and providing a better target for the agent.

## 5 EXPERIMENTS AND RESULTS

**Experiments setting** We evaluated DQfDD-BC in the classic OpenAI Gym environments: CartPole-v0, CartPole-v1, Acrobot-v1 and LunarLander-v2 (Brockman et al., 2016). We used the original reward functions provided by the gym environments without giving any additional rewards or penalties during the learning process. At the end of each episode, the rewards for all steps in the episode are summed up to form an episode score. The classical DDQN (Hasselt et al., 2016) and DQfD (Hester et al., 2018) algorithms are chosen as the benchmark algorithms to compare with the proposed approach.

Two experimental conditions were set up: perfect demonstrations and imperfect demonstrations. All demonstrations were obtained from pre-trained DDQN models and the major difference between perfect and imperfect demonstrations is as follows. The perfect demonstrations are obtained from state transitions data of a well-trained DDQN agent, while the imperfect demonstrations are derived from the model during the training process. In all demonstrations, the initial amount of transitions is around ten thousand before the training and the number will increase when new transitions are added into the demonstrations. The average episode scores of perfect demonstration collected from CartPole-v0, CartPole-v1, Acrobot-v1, LunarLander-v2 are 200, 500, -84, 246. and they are 357, -238, 14 in imperfect demonstrations of CartPole-v1, Acrobot-v1, LunarLander-v2. More implementation and experimental details, such as hyperparameters, termination conditions are presented in the appendix.

**Experimental results** Table 1 shows the comparison between DDQN, DQfD and DQfDD-BC in different gym environments. The average stop episode means the average stop episode number among different runs and roughly represents the learning speed and the average final score illustrates the final performance of these methods.

Table 1: Performance of different methods in different OpenAI Gym environments

| Env | DDQN | Perfect Demonstration | | Imperfect demonstration | |
|---|---|---|---|---|---|
| | | DQfD | DQfDD-BC | DQfD | DQfDD-BC |
| | | Average stop episode | | | |
| CartPole-v0 | 310±85 | 162±170 | **161**±170 | - | - |
| CartPole-v1 | 500±0 | 121±19 | **91**±27 | 500±0 | **411**±179 |
| Acrobot-v1 | 198±123 | 51±4 | **49**±5 | 261±136 | **143**±11 |
| LunarLander-v2 | 263±21 | 65±3 | **65**±9 | 402±192 | **119**±17 |
| Env | | Average training score for last 10 episodes | | | |
| CartPole-v0 | 193±0.6 | 190±10.7 | **194**±9.7 | - | - |
| CartPole-v1 | 133±12.1 | 498±3.9 | **500**±0.3 | 348±25.1 | **429**±46.0 |
| Acrobot-v1 | -97±5.3 | -95±4.0 | **-91**±2.5 | -96±5.3 | **-95**±7.8 |
| LunarLander-v2 | 231±6.5 | 219±8.7 | **223**±6.1 | **234**±4.4 | 211±20.4 |

In all environments, our DQfDD-BC method requires the smallest number of episodes to stop training and obtains almost the highest final performance, which demonstrates the significant advantages

of learning speed and performance. It needs to be pointed out that the termination condition of CartPole-v0 is much easier to be satisfied compared to the CartPole-v1. In some lucky episodes, the termination condition can be triggered in CartPole-v0, which is almost impossible in CartPole-v1. Thus the imperfect demonstration of CartPole-v0 is hard to be distinguished from the random policy and trained policy. To make the results more persuasive, we only test the proposed method in CartPole-v1 with the imperfect demonstration.

## 5.1 DQFDD-BC WITH PERFECT DEMONSTRATION

Compared with the DQfD algorithm, the proposed DQfDD-BC algorithm achieved better results in terms of convergence speed and learning capability. Fig. 2 shows the episode rewards of various methods in three gym environments. In CartPole-v0, the DQfDD-BC approach has a slightly faster convergence speed than DQfD during the early learning stage. Although some runs of DQfDD-BC and DQfD failed to satisfy the termination condition, the table 1 shows that the DQfDD-BC has a smaller average stop episode and it also gained a small advantage in final performance score. In CartPole-v1 and LunarLander-v2 environments, the DQfDD-BC shows obvious faster-learning speed and fewer episodes are needed for DQfDD-BC to get a higher score than DQfD.

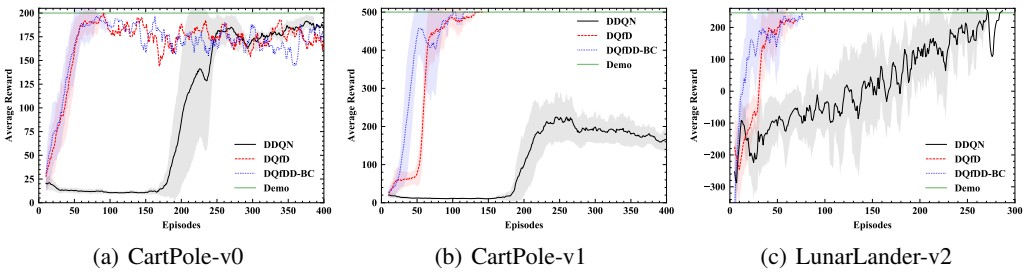

| (a) CartPole-v0 | (b) CartPole-v1 | (c) LunarLander-v2 |

Figure 2: The training episode scores of different methods with perfect demonstrations in different environments. The green line is the average scores of the perfect demonstrations.

Although the DDQN failed to trigger the termination condition before the experiment was forced to stop in CartPole-v1, both DQfD and DQfDD-BC can obtain a full score, which proves that the demonstration improves the learning ability of DRL agent. The DQfDD-BC algorithm retains the advantages of the DQfD algorithm and further improves the supervised loss function so that all self-generated data can obtain supervised losses, thus improving the learning speed.

## 5.2 DQFDD-BC WITH IMPERFECT DEMONSTRATIONS

We also validated the adaptability of the DQfDD-BC method through experiments with imperfect demonstrations. Since perfect demonstrations are difficult to define and acquire in many scenarios, it is more relevant to train the agent with imperfect demonstrations which are relatively better than the random policy. For example, the off-line optimized method can be used to generated imperfect demonstration. Fig. 3 (a) shows the episode scores of different method with imperfect demonstrations in the CartPole-v1 environment. The results show that the DQfDD-BC method outperforms DQfD by a significant margin in terms of convergence speed. By keeping the demonstrations updated, the proposed DQfDD-BC method allows the newly updated demonstration to avoid the negative influence brought by the imperfect demonstration and eventually converge to a better performance level.

The proposed DQfDD-BC method can adapt to imperfect demonstrations well and obtain higher average final performance than DQfD with imperfect demonstration. DQfD method shows great dependence on the demonstration and ultimately converges to the vicinity of the demonstration and imperfect demonstrations harm the performance of the DQfD method. We further explored the variation of each component of the complete loss function to figure out the reason why DQfDD-BC is able to get superior performance. Fig. 3 (b) shows that the TD losses (TD-1, TD-n) of DQfD have been stabilized within a small fluctuation range in the later stages of self-learning but the expert loss fluctuating over a wide range. This is partly due to the reason that the action output by the learned

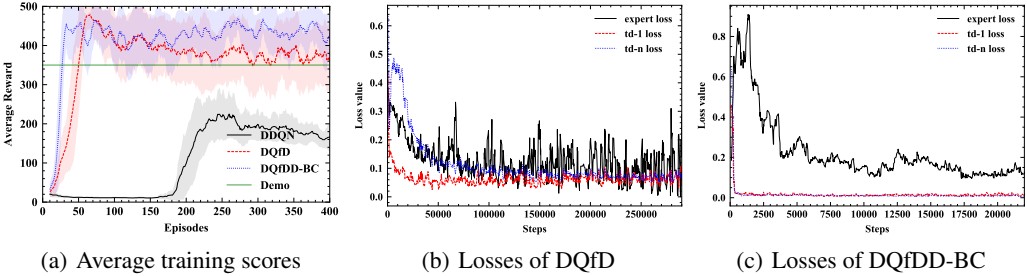

(a) Average training scores     (b) Losses of DQfD     (c) Losses of DQfDD-BC

Figure 3: Experimental result with imperfect demonstration. (a) The training score of different methods in the CartPole-v1 environment. The green line is the average scores of the demonstrations. (b) and (c) show the loss function value during the training process.

policy can be quite different from those sampled from the imperfect demonstrations. The agent can obtain a better policy through the TD losses and provides an appropriate action. However, the supervised loss draws the learned policy near the policy represented by the imperfect demonstration and inapposite actions are provided.

Additionally, the fixed imperfect demonstration is not immune to its own negative effects, resulting in the inability to converge to the highest score. Fig. 3 (c) illustrates that the expert loss of DQfDD-BC decreases more steadily and the agent achieves better final performance. The dynamic demonstration has the ability to modify the disadvantage of imperfect demonstration and expand the state-action space covered by the demonstration.

## 5.3 ABLATIONS

To figure out the importance of dynamic demonstrations and expert loss with BC model in the proposed DQfDD-BC method, we also test the method without dynamic demonstration or BC model in the CartPole-v1 environment. Fig. 4 shows the effect of integrating dynamic demonstrations and BC model to the original DQfD model. The figure clearly shows that when the dynamic demonstration is added to the DQfD algorithm (DQfDD), the model present a significant increase in training speed during the early stage. The DQfDD can also converge to the highest score and ended the training process in a short time while the DQfD needs some more time. This indicates that by adding new samples to the demonstrations, a large number of high-performance transitions can improve the generalization ability and training stability of the model.

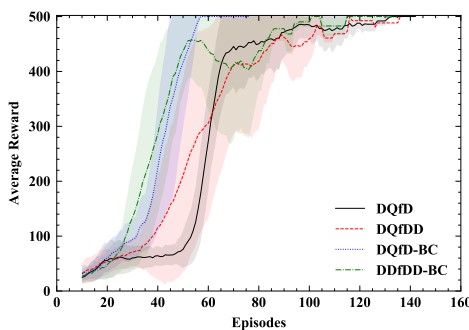

Figure 4: Ablation experimental results. The dynamic demonstration (DQfDD) and expert loss with a behavioral cloning model (DQfD-BC) are added to the original DQfD method.

In the case where the large margin expert loss is replaced by the loss generated by the BC model (DQfD-BC), the training speed of the model is significantly improved. Fig. 4 shows that the average training score of the DQfD algorithm reaches only about 80% of the convergence state, while the DQfD-BC model finishes the training process. The DQfD-BC has a slightly lower learning speed than the DQfDD-BC method during the learning process but the DQfD-BC ends the learning process earlier. By introducing a supervised model to generate an expert loss, the learning speed is improved significantly. Since the self-generated transitions can also be utilized to generate TD losses and expert loss with the supervised model, higher sample efficiency is achieved. The final DQfDD-BC model, which combines dynamic demonstrations and the expert loss with the BC model, simultaneously improves the learning speed and exhibits excellent performance.

## 6 CONCLUSION

In this work, we present an approach, namely the deep Q learning from dynamic demonstration integrated with behavioral cloning (DQfDD-BC), to enhance the learning efficiency and quality of DQfD. The experimental results demonstrate that regardless of the degree of perfection of historical demonstrations, DQfDD-BC can effectively improve the learning speed and final performance compared with DQfD in generic environments. From an ablation experiment, both the dynamic demonstration and expert loss with the BC model showed their own potential to improve the learning quality.

As our approach provides a feasible solution to bridge the DRL methods and the decision-making problems in complex real-world environments, we will later apply our model to real-world settings, such as a large-scale urban traffic control system. While considering the process of recording human demonstrations, we will explore and improve our model for the ability to handle engineering issues in data sparsity, human diversity, and so on.

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

## A  APPENDIX

**Hyperparameters** There are numerous hyperparameters in the experiments and different hyperparameters have a significant effect on the results. In all experiments, the $n$ in $n - step$ return is 10, the batch_size = 64, the buffer_size=1000000, the learning_rate=0.00001, the learning rate of supervised model is 0.001, and the discount factor $\gamma = 0.95$ in the CartPole environments (0.99 for other environments). We find that different random seeds have different performances and we choose $[100, 200, 300, ..., 1000]$ as the random seed list. In the $\epsilon - greedy$ policy, $\epsilon$ linearly decreases from a maximum value of $0.99$ to a minimum value of $0.001$ and remains at the minimum value. In the CartPole-v1 environment, the descent is maintained for 10000 interaction steps and in other environments 5000. In the prioritized experience replay buffers, new transitions are given the highest probability of being sampled and are later updated based on the absolute values of TD-1 errors. The parameter $\beta$ controls the amount of importance sampling and it is annealed linearly from $\beta_0$ to 1. In DQfD and DQfDD-BC, $\lambda$ decides how to combine different losses. In the experiments, $\lambda_1 = \lambda_2 = 0.5$, $\lambda_3 = 0.001$ and the L2 regularization is provided by the optimizer in the PyTorch. The large margin in DQfD. is fine-tuned and set to be 1. A margin that is too large will cause the model to rely heavily on the supervised learning process and lose its ability to learn independently, while a margin that is too small will cause the advantages of supervised learning to be unused.

**Model architecture** Since the state of experiment environments are vectors, fully-connected layers are appropriate for the requirements and they are leveraged to construct the Q network and the BC model. In the BC model, the hidden unit number of all three fully-connected layers are $[64, 24, 12]$

and $LeakyRelu$ is selected as the activation function. The Q network which has $[512, 512, 256, 128]$ hidden units in every hidden layer and the activation function is $Relu$.

**Termination condition** In the experiments, the termination condition was as following. Within 1000 episodes (it is 500 in CartPole-v0 and v1), the average episode score of last 30 consecutive episodes is greater than certain scores (CartPole-v0: 190, CartPole-v1: 490, Acrobot-v1: -100, LunarLander-v2: 200). The models were considered to be successfully converged if the termination conditions are satisfied. Each method is repeated 10 times (it is 5 in CartPole-v0 and v1) with different random seeds.

