# OpenReview forum: "Deep Q Learning from Dynamic Demonstration with Behavioral Cloning"
_ICLR.cc/2021/Conference — Reject_

### Official Review · AnonReviewer2 · 2020-10-20

**Rating:** 7
**Confidence:** 4

**Review:**

=====POST-REBUTTAL COMMENTS========

I thank the authors for the response and the efforts in the updated draft. Most of my concerns were addressed. This is a simple, but nice idea. After reading the rebuttal and the other reviews I am recommending to accept the paper.

##########################################################################

Summary:


This paper seeks to improve deep reinforcement learning from demonstrations via a supervised learning loss based on behavioral cloning. The results show that adding the BC loss stabilizes training and shows improvements over baselines for OpenAI Gym environments.

##########################################################################

Reasons for score:


I think the idea is nice, but it is hard to follow as written and the experiments are on simple problems. It would be better to compare against more complex benchmarks such as Atari. The writing needs improvement for clarity.


##########################################################################
Pros:


1. The ablation study in Figure 4 shows the benefit of the proposed method.

2. The ability to outperform a suboptimal demonstrator is important for imitation learning in the real world.




##########################################################################

Cons:

1. The idea of adding new trajectories from high scores was interesting, but I was left wondering why this works. Won't this add a lot of really bad trajectories in the beginning of learning which seem like they would have a negative impact on the BC policy. Also, the function "get the best episode score" was never rigorously defined.

2. Equations should be introduced before they are included. Equations 1-4 come before the text that talks about them and this makes the paper more confusing that it needs to be.

3. Equation (5) seems incomplete. Shouldn't l(\pi, \pi_bc, s_t) be a function of the action? Otherwise it is just a constant per state, but it is in the max_a brackets. Also it is unclear what the difference between s_t and s are in Eqn (5).

4. Average stop episode is not clearly defined in the text. In the appendix it explains more but the thresholds are not justified.

---

> ### Author Response · Authors · 2020-11-23
> **Reply**
>
> Comment 4.1
>
> I think the idea is nice, but it is hard to follow as written and the experiments are on simple problems. It would be better to compare against more complex benchmarks such as Atari. The writing needs improvement for clarity.
>
> **Feedback 4.1**:
>
> Thanks for your advice. We tried our best to compare against more complex benchmarks but the experiments have not finished before we submit the updated manuscripts. And we have polished the English writing of the paper.
>
>
> Comment 4.2:
>
> 1. The idea of adding new trajectories from high scores was interesting, but I was left wondering why this works.
> 2. Won't this add a lot of really bad trajectories in the beginning of learning which seem like they would have a negative impact on the BC policy.
> 3. Also, the function "get the best episode score" was never rigorously defined.
> 4. Equations should be introduced before they are included. Equations 1-4 come before the text that talks about them and this makes the paper more confusing that it needs to be.
> 5. Equation (5) seems incomplete. Shouldn't l(\pi, \pi_bc, s_t) be a function of the action? Otherwise it is just a constant per state, but it is in the max_a brackets. Also it is unclear what the difference between s_t and s are in Eqn (5).
> 6. Average stop episode is not clearly defined in the text. In the appendix it explains more but the thresholds are not justified.
>
> **Feedback 4.2**:
>
> 1. Trajectories with high scores own a greater probability of containing some transitions that successfully deal with difficult scenarios, which can not only expand the state-action space covered by the transition samples in the experience replay buffer but also help the supervised learning model provide more effective information to guide the convergence of the model.
> 2. We have also noticed this feature and to reduce the negative influence as much as possible, only a small number of samples are added to the demonstration during the early stage of the learning and more volume of samples are generated after the model has gained better performance.
> 3. The best episode score is taken from the final cumulative reward of each episode compared to the previous history. To reduce the influence of the lucky episode, we don’t update the demonstration in the early stage of the self-learning process. In the experiments, we begin to update the demonstration after the self-learning process has last for at least 10 episodes.
> 4. Thanks for your suggestion, we will improve the layout to make the paper clearer.
> 5. $l(\pi, \pi_{bc};s_t)$ is indeed a function of action and it determines the loss value based on the difference between $\pi_{bc}(s)$ and other actions in the action space. We have modified the equation, which looks like $l(a, \pi_{bc}(s))$, to clarify the meaning. $s_t$ is the same as s and we have made them consistent.
> 6. The average stop episode is the average value of the stop episode number among different runs. Because of the randomness in the experiments and environments, the stop episode number (after which the termination condition is satisfied) of different runs varies from each other. The threshold in the termination condition is equal to or higher than the *rThresh*, which is a reward threshold. If the agent gets a higher score than the *rThresh*, the task can be considered as solved.

---

### Official Review · AnonReviewer3 · 2020-10-22
**Tend to accept**

**Rating:** 6
**Confidence:** 2

**Review:**

Summary:

This paper proposes integrating deep Q-learning from dynamic demonstrations with a behavioral cloning model (DQfDD-BC). Compared with DQfD, the proposed approach introduces a behavior cloning model, which was first pre-trained by leveraging historical demonstrations and then updated using generated dynamic demonstration. The BC model is used in the expert loss function, where the DRL model's actions are compared with those obtained from the BC model for policy improvement guidance. The experimental results in OpenAI Gym environments show that the proposed approach adapts well to different demonstrations' imperfection levels and accelerates the learning processes. The ablation study also indicates that the new method improves the learning convergence performance compared with the original DQfD model.

========================

Reasons for score:
Overall, based on the description of the paper, the proposed approach works well.

========================

Detailed comments:
- In the proposed approach, the BC model is fine-tuned if the model achieves a relative high-performance score. With this mechanism,  high-quality transition samples are included, and it helps avoid adverse impacts caused by imperfect demonstrations.
- The experiments show that the performance of DQfDD-BC is better than DQfQ, and it works well for imperfect demonstration scenarios.
- In this paper, I think the primary idea is about how to leverage high-quality data in the generated trajectory well. This paper aims to build a BC model, initialized by demonstration dataset, and later finetuned with newly generated high quality data. One thing I do not understand well is the necessity of building the BC model. What if we use some heuristic approaches to replace the replay buffer with higher quality data? Can some simple strategies like this achieve similar performance as building a BC model?

========================

Some obvious typos:
- Section 5 Experimental results: environmets -> environments
-Section 5.3, paragraph 2: combins -> combines

---

> ### Author Response · Authors · 2020-11-23
> **Reply**
>
> Comment 3.1:
> One thing I do not understand well is the necessity of building the BC model. What if we use some heuristic approaches to replace the replay buffer with higher quality data? Can some simple strategies like this achieve similar performance as building a BC model?
>
> **Feedback 3.1**:
> Thank you for the suggestion. Heuristic methods usually contain more prior knowledge, resulting in that the samples selected may be biased. In this paper, the BC model was selected is mainly due to its advantage of the characteristics of fast training of supervised learning, and it can also provide decision-making results which have totally different mechanism from reinforcement learning. We clarify the advantages of BC in the revised manuscript to support our method.
>
> Comment 3.2:
> Some obvious typos:
> Section 5 Experimental results: environmets -> environments
> Section 5.3, paragraph 2: combins -> combines
>
> **Feedback 3.2**:
> Thanks for your careful review and we have polished the manuscript carefully to make the paper better.

---

### Official Review · AnonReviewer1 · 2020-10-27
**Official Blind Review**

**Rating:** 6
**Confidence:** 4

**Review:**

Summary

This paper proposes a new approach (DQfDD-BC) to leverage demonstrations in the framework of deep Q-learning. The new approach augments the previous approach called DQfD by
a) adding self-generated trajectories with high rewards as additional demonstrations (a feature the authors called “dynamic demonstrations”),
b) replacing large-margin loss with a cross-entropy loss between the learner agent and a helper policy learned by behavior cloning demonstrations.
The new approach improves upon the previous by better leveraging demonstrations and being more robust against imperfect demonstrations. The authors demonstrated these improvements in experiments by showing the new approach leads to accelerated training speed (in terms of episodes) when using either perfect or imperfect demonstrations, and higher performance (in terms of rewards) when using imperfect demonstrations.

==============

Positives

This paper is well motivated. The two intended benefits of the new approach are meaningful and important.

The experiments back up the authors’ claim on the benefits, especially on the improvement of training speed. In the CartPole tasks, the training speed is improved by 2 folds.

The paper is written clearly. It was easy to understand the new approach and the differences on top of the previous one.

==============

Negatives

The extensiveness of experiments is underwhelming, both in the diversity and complexity of the tasks chosen. The four tasks are among the most simple ones typically used for comparable algorithms, having very low degrees of freedom and very simple dynamics. For example, DQfD was validated on a broader and more challenging set of Atari tasks, including Montezuma Revenge where (human) demonstrations are critical for a learner agent to tackle the task.

By introducing a BC-learned helper policy, the approach introduces dependencies on how well this helper policy is (in terms of approximating the demonstrations) in order to provide better loss signals to large-margin loss. It’d be good to design additional experiments to showcase shortcomings, if any, of this proposed approach. For example, the authors used ten thousands transitions as initial demonstrations. It’d be interesting to see how the proposed approach would fare when different amounts of demonstrations are available.

==============

Recommendation

Overall the approach proposed is interesting and promising. I’d recommend a weak accept. The authors are strongly encouraged to further improve the experiments.

==============

Minor Comments

Table 1 is confusing to read. Especially in the top half, “Demo” rewards are juxtaposed with other numbers in a completely different units (number of episodes).

There are some typos. In section 2, it’s not clear what determination is meant in “determination of DAGGER”. In Section 1 “difficult consistent” -> “difficult to be consistent”.

---

> ### Author Response · Authors · 2020-11-23
> **Reply**
>
> Comment 2.1:
>
> The extensiveness of experiments is underwhelming, both in the diversity and complexity of the tasks chosen. The four tasks are among the most simple ones typically used for comparable algorithms, having very low degrees of freedom and very simple dynamics. For example, DQfD was validated on a broader and more challenging set of Atari tasks, including Montezuma Revenge where (human) demonstrations are critical for a learner agent to tackle the task.
>
> By introducing a BC-learned helper policy, the approach introduces dependencies on how well this helper policy is (in terms of approximating the demonstrations) in order to provide better loss signals to large-margin loss. It’d be good to design additional experiments to showcase shortcomings, if any, of this proposed approach. For example, the authors used ten thousands transitions as initial demonstrations. It’d be interesting to see how the proposed approach would fare when different amounts of demonstrations are available.
>
> **Feedback 2.1**:
>
> Thank you for the advice. We will try our best to test our method on broader and more challenging environments. The influence of the number of demonstrations also deserves more research work.
>
> Comment 2.2:
>
> Table 1 is confusing to read. Especially in the top half, “Demo” rewards are juxtaposed with other numbers in completely different units (number of episodes).
>
> There are some typos. In section 2, it’s not clear what determination is meant in “determination of DAGGER”. In Section 1 “difficult consistent” -> “difficult to be consistent”.
>
> **Feedback 2.2**:
>
> Thanks for your careful review and we have revised the manuscripts to make it clearer.

---

### Official Review · AnonReviewer4 · 2020-10-28

**Rating:** 5
**Confidence:** 5

**Review:**

This paper is introducing a learning method which combines both Imitation Learning and Reinforcement Learning, such that an autonomous learner can leverage prerecorded expert knowledge. In comparison to previous work, this model has an expert cost function which gives priority to the expert behavior, not only using the expert demos (like in DQfD), but also with a model trained with those demos using  behavioral cloning, such that it could be evaluated in states that were not visited during the demonstrations. Additionally, new executions of the learner that have high performance are included in the buffer for training the policy imitating the expert, since they can also be considered new better demonstrations.

The paper presents an interesting idea with potential, although there are some aspects to consider mainly about the presentation of the paper, in order to improve its content, as listed below. However, the aspects that are not clear in the current version of the paper need to be fixed before considering acceptance, since there are clear open questions.

Section 2 Related work needs some improvement:
- First sentence of Section 2 could be improved for it to be more meaningful "Learning from demonstration (LfD) is a class of decision-making methods by learning from demonstration and  imitation  learning is a subset of LfD", additionally LfD are not exactly decision-making methods.
- It is stated that BC received extensive attention "due to its great performance", I'd say rather "due to its simplicity", since it is simply the application of supervised learning, there is extensive literature arguing that it does not always perform well, therefore several other methods have been proposed.
- "Behavioral cloning is an end-to-end learning method" is not a right statement.
- Authors mention about DAgger that "... address the shortcoming of limited state-action space covered by demonstration in traditional imitation learning". This is an indirect result, but actually it is intended to cope with the covariate shift problem.
- in "...adversarial learning method and has yielded impressive results...", impressive could be replaced by an objective word.

Regarding section 4:- In "As the transition quality of historical demonstrations is usually much higher than that associated with stochastic policies..." stochastic policies are not necessarily something meaningless, I guess the authors meant a completely random policy.
- In the line 20 of Algorithm 1. how the "get the best episode score" is computed? it is just taken from the final cumulative reward of each episode compared to the previous history? if that's the case how to deal with lucky episodes in which the conditions are simply easier for the agent, and not necessarily that the current policy is indeed better.
- It's not so clear why the self learning process is executed, my interpretation is that authors use it to update the BC policy towards \pi(s), such that suboptimal behaviors in the demonstrations don't keep pulling the learning policy.
- lines 8 and 18 have different notations for the same operation,  authors could use only one of them to be more consistent.
Regarding the experiments section- There are missing details of the experiments that do not allow for reproducing the procedure, how many repetitions were run for each case in Fig 2 and 3?, and also for the ablation study? It is not clear whether the results of Fig 2 and 3 are obtained from the same experiments that gave the results in Table 1, at least they do not seem to match. Why testing with the 2 variants of the CartPole environment instead of another kind of problem? since the difference between them is very simple

- Why is there no data in Table 1 for the cases of imperfect demonstrations for the CartPole-V0 environment?
- Why the learning curves of DDQN for the CartPole-V0 and CartPole-V1 cases are not similar at least during the first 200 episodes, looking at the mean and variance, both environments are not close to reach the maximum time steps, therefore in that region they should behave exactly the same.
- In Table 1, it is not so clear what average stop episode means, if it is the episode number in which the stop condition was reached, why are there negative numbers for the acrobot? if they are rewards obtained in the last episode, why are rewards higher than the possible to obtain in CartPole-V0.
- It is mentioned  "...but the DDQN and DQfD fail to meet the experimental termination condition with imperfect demonstration", However this is completely wrong, DDQN do not use demonstrations.

- It would be interesting to additionally mention how to define the size of the margin for the expert cost function, because this may depend on the kind of reward functions/range of value functions.

---

> ### Author Response · Authors · 2020-11-23
> **Reply-part2**
>
> Comment 1.7:
>
> 1. Regarding the experiments section- There are missing details of the experiments that do not allow for reproducing the procedure, how many repetitions were run for each case in Fig 2 and 3?, and also for the ablation study? It is not clear whether the results of Fig 2 and 3 are obtained from the same experiments that gave the results in Table 1, at least they do not seem to match. Why testing with the 2 variants of the CartPole environment instead of another kind of problem? since the difference between them is very simple
> 2. Why is there no data in Table 1 for the cases of imperfect demonstrations for the CartPole-V0 environment?
> 3. Why the learning curves of DDQN for the CartPole-V0 and CartPole-V1 cases are not similar at least during the first 200 episodes, looking at the mean and variance, both environments are not close to reach the maximum time steps, therefore in that region they should behave exactly the same.
> 4. In Table 1, it is not so clear what average stop episode means, if it is the episode number in which the stop condition was reached, why are there negative numbers for the acrobot? if they are rewards obtained in the last episode, why are rewards higher than the possible to obtain in CartPole-V0.
> 5. It is mentioned  "...but the DDQN and DQfD fail to meet the experimental termination condition with imperfect demonstration", However this is completely wrong, DDQN do not use demonstrations.
> 6. It would be interesting to additionally mention how to define the size of the margin for the expert cost function, because this may depend on the kind of reward functions/range of value functions.
>
> **Feedback 1.7**:
>
> 1. In all the experiments, the allowed maximum episodes are set to 1000 and 10 repetitions are completed to get the final results. Since different repetitions trigger the termination condition at different episodes, the data in the table are the statistic value (mean±std) and the data in the figure shows more detailed information about the learning process. The reason why testing with two CartPole environments is that the difficulty of the two environments is different. The maximum step of CartPole-v1 is 500 but it is 200 in CartPole-v0, more steps mean more carefully controlling are needed to satisfy the requirement of the environment. Thus, we complete different experiments on these two environments.
> 2. Compared to the Cartpole-v1, the termination condition of Cartpole-v0 is much easier to be satisfied. In some lucky episodes, the termination condition can be triggered in Cartpole-v0, which is almost impossible in Cartpole-v1. To make the results more persuasive, we only test the method in Cartpole-v1 and we add this reason in the revised manuscript.
> 3. Thanks for your comment, we have checked the experimental results and updated the results. We test our method in the CartPole environments again and the reward curves of DDQN are similar in newly obtained results. Due to time constraints, we only repeated the experiment 5 times and the maximum number of episodes was adjusted to 500 when we re-run the experiments. We also updated the ablation results.
> 4. In Table 1, the average stop episode means the average stop episode number among different runs. The data in the “Demo” columns is the average score and a negative number in an Acrobot environment is normal because the reward in Acrobot environment is negative. The demonstration score is evaluated through the episode score, so it is negative in Acrobot. We update the table to make it clear.
> 5. We have corrected the sentence to avoid the misunderstanding as follow: “…but DQfD fail to meet the experimental termination condition with imperfect demonstration and DDQN also failed to trigger the termination condition before the experiment was forced to stop”
> 6. The margin is a hyperparameter and it is fine-tuned in the experiments. A margin that is too large will cause the model to heavily rely on the supervised learning process and lose its ability to learn independently, while a margin that is too small will cause the advantages of supervised learning to be unused. In our experience, it has to be in the same order of magnitude and smaller as other loss items, so that it can have a more obvious impact on the complete loss while retaining a high self-learning ability.

---

> > ### Comment · AnonReviewer4 · 2020-11-24
> >
> > Thanks to the authors for replying to each of the points mentioned in the review.
> >
> > The feedback for comments 1.1 to 1.6 is fine.
> >
> > 1.7.1 The two versions of CartPole indeed have only that difference on the maximum duration of the episodes, but still, the observations, the actions, the dynamics, and the reward function are the same, so not much-added value with one of the two cases, which could have been replaced by another problem with different challenges.
> >
> > 1.7.2 "To make the results more persuasive" does not sound very objective, are results not shown if they are not good enough?
> >
> > 1.7.3 It is good you could add results from a higher number of runs, although I wonder what has changed because the new plots do not overlap with the previous ones considering the low variance, then it looks like they are obtained with different methods.
> >
> > 1.7.4 I notice the results have been also updated apart from the clarification of the table. I wonder why does it take longer for the CartPole-v0 than the v1 since as you also pointed out, the v1 is more challenging.
> >
> > 1.7.5 Ok.
> >
> > 1.7.6 This is nice to be included somewhere, perhaps in the supplementary material.

---

> ### Author Response · Authors · 2020-11-23
> **Reply-part1**
>
> Comment 1.1:
> 1. First sentence of Section 2 could be improved for it to be more meaningful "Learning from demonstration (LfD) is a class of decision-making methods by learning from demonstration and  imitation learning is a subset of LfD", additionally LfD is not exactly decision-making methods.
> 2. It is stated that BC received extensive attention "due to its great performance", I'd say rather "due to its simplicity", since it is simply the application of supervised learning, there is extensive literature arguing that it does not always perform well, therefore several other methods have been proposed.
>
> **Feedback 1.1**:
> 1. We agree with you that this sentence can be more meaningful and we checked the literature and found that LfD is a class of approaches to coping with different tasks quickly by learning from human or other intelligent methods. It is often used in robot control problems. To focus on the model proposed in the paper, we updated the manuscript in which there is no concept of LfD.
> 2. We partly agree with your idea that BC is simple and does not always perform well, but BC also has its own significant advantages, such as fast learning speed, simple and effective utilization of demonstration, and so on. We have updated the original text to make it more accurate.
>
> Comment 1.2:
> "Behavioral cloning is an end-to-end learning method" is not a right statement.
>
> **Feedback 1.2**:
> This statement is not precise enough, therefore we delete this sentence.
>
> Comment 1.3:
> Authors mention DAgger that "... address the shortcoming of limited state-action space covered by demonstration in traditional imitation learning". This is an indirect result, but actually, it is intended to cope with the covariate shift problem.
>
> **Feedback 1.3**:
> We agree with your statement that DAgger is proposed to cope with the covariate shift problem. In the Dagger algorithm, experts are required to respond to self-generated states. Thus, totally new transitions are generated and they can expand the state-action space covered by demonstration.
>
> Comment 1.4:
> in "...adversarial learning method and has yielded impressive results...", impressive could be replaced by an objective word.
>
> **Feedback 1.4**:
> We have updated the sentence, which becomes "… an adversarial learning method and has made efforts on  sparse and dense reward environments"
>
> Comment 1.5:
> Regarding section 4: In "As the transition quality of historical demonstrations is usually much higher than that associated with stochastic policies..." stochastic policies are not necessarily something meaningless, I guess the authors meant a completely random policy.
>
> **Feedback 1.5**:
> We have replaced the “stochastic policies” with “random policies”.
>
> Comment 1.6:
> 1. In the line 20 of Algorithm 1. how the "get the best episode score" is computed? it is just taken from the final cumulative reward of each episode compared to the previous history? if that's the case how to deal with lucky episodes in which the conditions are simply easier for the agent, and not necessarily that the current policy is indeed better.
> 2. It's not so clear why the self-learning process is executed, my interpretation is that authors use it to update the BC policy towards \pi(s), such that suboptimal behaviors in the demonstrations don't keep pulling the learning policy.
> 3. lines 8 and 18 have different notations for the same operation,  authors could use only one of them to be more consistent.
>
> **Feedback 1.6**:
> 1. The best episode score is taken from the final cumulative reward of each episode compared to the previous history. To reduce the influence of the lucky episode, we do not update the demonstration in the early stage of the self-learning process. In the experiments, we begin to update the demonstration after the self-learning process has last for 10 episodes.
> 2. The reason is indeed the same as your interpretation. We have clarified the significance of the self-learning process. The supervised learning process and self-learning process can promote each other. The supervised model provides a basic reference and the self-learning process overcomes the negative influence on the BC model brought by the suboptimal samples.
> 3. Thanks for your advice and we have updated the manuscript.

---

### Author Response · Authors · 2020-11-23
**Update the manuscript**

We thank all the reviewers for their valuable comments and advice. Please find the uploaded revised manuscript. We revised the manuscript according to the reviewers' comments to address concerns and to avoid any unnecessary confusion. All changes in the revised manuscript are highlighted in blue.

---

### Decision · Program_Chairs · 2021-01-07
**Final Decision**

**Decision:**

Reject

**Comment:**

The reviewer acknowledged that the proposed method is simple and seems to work well on the chosen benchmarks. Yet the expressed several concerns that were not fully addressed by the authors in their responses. The major concern is about the experimental setup. The chosen tasks have been judged too simple and quite different from those where the baselines were tested initially (e.g. DQfD was demonstrated on a diverse set of Atari games).

The clarity of the paper should also be improved. For instance, the way the number of trajectories that are added to the replay buffer increases with time is not well explained and it seems to be crucial for the algorithm to outperform the baselines. The authors also seemed to select the experiments so that the "results are more persuasive", discarding experiments where you can be unlucky. This looks very much like cherry-picking and didn't convince the reviewers.